# Neural network and layer-wise relevance propagation reveal how ice hockey protective equipment restricts players' motion

**Rebecca Lennartz** [1,2]*, **Arash Khassetarash**[2], **Sandro R. Nigg**[2], **Bjoern M. Eskofier**[1,3], **Benno M. Nigg**[2]

**1** Machine Learning and Data Analytics Lab, Department Artificial Intelligence in Biomedical Engineering (AIBE), Friedrich-Alexander-Universität Erlangen-Nürnberg (FAU), Erlangen, Germany, **2** Human Performance Laboratory, Faculty of Kinesiology, University of Calgary, Calgary, Alberta, Canada, **3** Translational Digital Health Group, Institute of AI for Health, Helmholtz Zentrum München—German Research Center for Environmental Health, Neuherberg, Germany

\* rebecca.lennartz@fau.de

**Data Availability Statement:** All relevant data are within the manuscript and its Supporting Information files.

**Funding:** RL was supported by a fellowship of the German Academic Exchange Service (DAAD) by

## Abstract

Understanding the athlete's movements and the restrictions incurred by protective equipment is crucial for improving the equipment and subsequently, the athlete's performance. The task of equipment improvement is especially challenging in sports including advanced manoeuvres such as ice hockey and requires a holistic approach guiding the researcher's attention toward the *right* variables. The purposes of this study were (a) to quantify the effects of protective equipment in ice hockey on player's performance and (b) to identify the restrictions incurred by it. Twenty male hockey players performed four different drills with and without protective equipment while their performance was quantified. A neural network accompanied by layer-wise relevance propagation was applied to the 3D kinematic data to identify variables and time points that were most relevant for the neural network to distinguish between the equipment and no equipment condition, and therefore presumable result from restrictions incurred by the protective equipment. The study indicated that wearing the protective equipment, significantly reduced performance. Further, using the 3D kinematics, an artificial neural network could accurately distinguish between the movements performed with and without the equipment. The variables contributing the most to distinguishing between the equipment conditions were related to the upper extremities and movements in the sagittal plane. The presented methodology consisting of artificial neural networks and layer-wise relevance propagation contributed to insights without prior knowledge of how and to which extent joint angles are affected in complex maneuvers in ice hockey in the presence of protective equipment. It was shown that changes to the equipment should support the flexion movements of the knee and hip and should allow players to keep their upper extremities closer to the torso.

the Federal Ministry of Education and Research of Germany (BMBF) (no grant number available). The funders had no role in study design, data collection and analysis, decision to publish, or preparation of the manuscript. Data collection was funded by CCM Hockey (Montreal, Canada) and Biomechanigg Sport and Health Research (BSHR; Calgary, Canada). CCM Hockey also funded subject recruitment and data collection. However, the results presented in this article do not in any way represent a bias towards CCM Hockey over other brands. The results of the study are also presented clearly, honestly, and without fabrication, falsification, or inappropriate data manipulation. The funders had no role in study design, data collection and analysis, decision to publish, or preparation of the manuscript.

**Competing interests:** This work was funded by CCM Hockey (Montreal, Canada) and Biomechanigg Sport and Health Research (BSHR; Calgary, Canada). CCM Hockey also funded subject recruitment and data collection. However, the results presented in this article do not in any way represent a bias toward CCM products over other brands. The principal investigator, Dr. Benno Nigg, is also the Chief Science Officer of the sponsoring company BSHR. BSHR covered material costs for this research and was simply interested in the outcome of the study, regardless of the findings. The company BSHR did not benefit from the results of the findings. BSHR had no influence on the outcome of this study. The results of the study were presented clearly, honestly, and without fabrication, falsification, or inappropriate data manipulation. The funders had no role in study design, data collection and analysis, decision to publish, or preparation of the manuscript. This does not alter our adherence to PLOS ONE policies on sharing data and materials.

## Introduction

Ice hockey is a fast-paced game, requiring a wide variety of movements ranging from sprints and rapid deceleration and acceleration around tight turns to high-speed collisions with other players. Hence, ice hockey injury rates are among the highest in competitive sports [1]. Besides the introduction of several initiatives such as behavioral modifications and changes in rules [2], the use of protective equipment has been shown to decrease the rates of injuries [3]. On the other hand, the use of heavy, cumbersome, and restrictive protective equipment inevitably causes movement limitations and may affect the natural movement patterns of hockey players.

Although protective equipment has been evaluated for injury reduction across various studies [3–5], there is a lack of literature concentrating on the performance aspect of protective equipment. Existing research investigating performance in ice hockey has mostly focused on the skates and the stick. For example, it has been shown that the design of skate blades can improve the skating speed of players by up to 1.3% [6], and the stiffness of a hockey stick shaft can significantly influence the puck velocity in wrist shots [7]. Regarding the protective equipment, previous work only focused on the reduction of weight to decrease sprint times and increase endurance [8]. Besides the added weight, the effects of protective equipment on the kinematics of movements are currently unknown.

The key technical challenge in understanding the effects of equipment on skating technique is analyzing these data to identify when and where kinematic differences exist. Conventional biomechanical analysis methods addressed the problem of high dimensionality data by focusing on single, time-discrete variables that are pre-selected by the researcher, for example, peak impact force in running [9, 10]. However, finding reliable and representative variables is still controversial in well-investigated sports such as running [11–13] and it is even more difficult in sports and movements less understood and investigated, such as ice hockey. Therefore, researchers are at risk of discarding potentially relevant data while omitting temporal information and time dependencies. Further, traditional correlations and statistical hypothesis tests (e.g., t-tests, ANOVAs) for single points do not extend to curves [12, 14]. In response to these shortcomings of conventional statistical tools, machine learning techniques have become popular to help deal with the increasing amount of data in the biomechanical research [15]. Especially artificial neural networks, ANNs, have been used in biomechanical contexts, for instance, to identify subject-specific locomotion characteristics [16] or to distinguish between different activities in badminton [17].

Although artificial neural networks, are successfully applied in biomechanics for classification tasks, they introduced another challenge based on their black box characteristic. Due to the structure of an artificial neural network, it is impossible to understand *how* the machine learning algorithm came to its decision [18]. Consequently, artificial neural networks are limited to the result and provide only limited information about the relationship between the input data and a particular (classification) result. Thus, the question to be answered in this study is not whether the movement was performed with equipment or not, but *how* this decision was made based on the player's range of motion.

In this context, multiple procedures and algorithms [19–21] have been proposed to overcome the black-box problem and to improve the usability of artificial neural networks. These powerful tools and therefore has raised attention across different disciplines. One of these methods is the layer-wise relevance propagation, developed by Bach and colleagues [22], which has been successfully applied to several biomechanical questions such as understanding the uniqueness of human locomotion in walking [23], and running [24]. The basic concept of layer-wise relevance propagation (LRP) is based on the assignment of relevance scores to the different input variables dependent on their contribution to the model's decision. Input

variables that strongly influence the model's decision have higher relevance scores than variables that have a weak influence on the model's decision, and therefore lower relevance scores. Therefore, the implementation of layer-wise relevance propagation in combination with machine learning techniques can resolve the black-box limitation, and lead the researcher's focus toward the most relevant variables [22].

In this study, we utilized artificial neural networks along with layer-wise relevance propagation methods to investigate the effects of hockey protective equipment on player's performance to identify key biomechanical changes. We hypothesized that the protective equipment would restrict the player's motion in a way that would be detectable by a neural network. Specifically, the purposes of this work were to:

1. Train neural network models for a series of movements to distinguish between the presence of protective equipment and the absence of protective equipment while performing ice hockey drills based on their joint angles captured by an inertial measurement unit-based system.

2. Identify the variables and intervals during each of the movements that were most relevant in distinguishing between the two equipment conditions using layer-wise relevance propagation.

## Methods

### Participants

Twenty male ice hockey players (age: 26.4 ± 7.0 years; mass 86.4 ± 6.5 kg; height 183.8 ± 6.5 cm) participated in this study (06. April– 22. June 2022). To ensure comparable effect of the protective equipment, participation was limited to active male ice hockey players. Therefore, all participants were active male ice hockey players, including 14 elite players and 6 recreational players. Written informed consent was collected from all participants and approval for this research project was obtained from the ethics board of the University of Calgary (Ethics ID: REB21-1932).

### Data collection

The participants completed four different drills to gain information on different complex maneuvers: A sprint drill, a linear crossover drill, a power turn drill, and a slap shot. These drills were chosen to represent a variety of movements required in an ice hockey game [25]. For the sprint drill, the participants skated the long axis of the ice rink as fast as possible. To record the sprint time, timing lights (Brower TCi Timing System by Brower Timing Systems, Draper, Utah, USA), were placed 20 m apart. For the linear crossover drill, participants skated the same distance, while performing the linear crossover movement instead of sprint strides for stroking. The slap shot was performed from a mark of 5 m from the goal line centered in front of the goal. The players were instructed to shoot as strongly as possible with their preferred side. A radar gun (Stalker Sport 2, Stalker sport, Richardson, Texas, USA) was positioned behind the goal to capture the shooting velocity of the puck. For the power turn drill, the participants skated back and forth between two face-off dots on the ice. Reaching one mark, players had to perform a power turn (i.e., sharp turn) around the mark and skate back to the first mark, where they completed a second power turn in the other direction, creating a figure-eight pattern. The circuit was repeated three times. Illustrations of the executed drills are displayed in Fig 1.

The drills were performed successively two times per condition in the following order: power turn, linear crossover, slap shot, and sprint. The participants were advised to take as

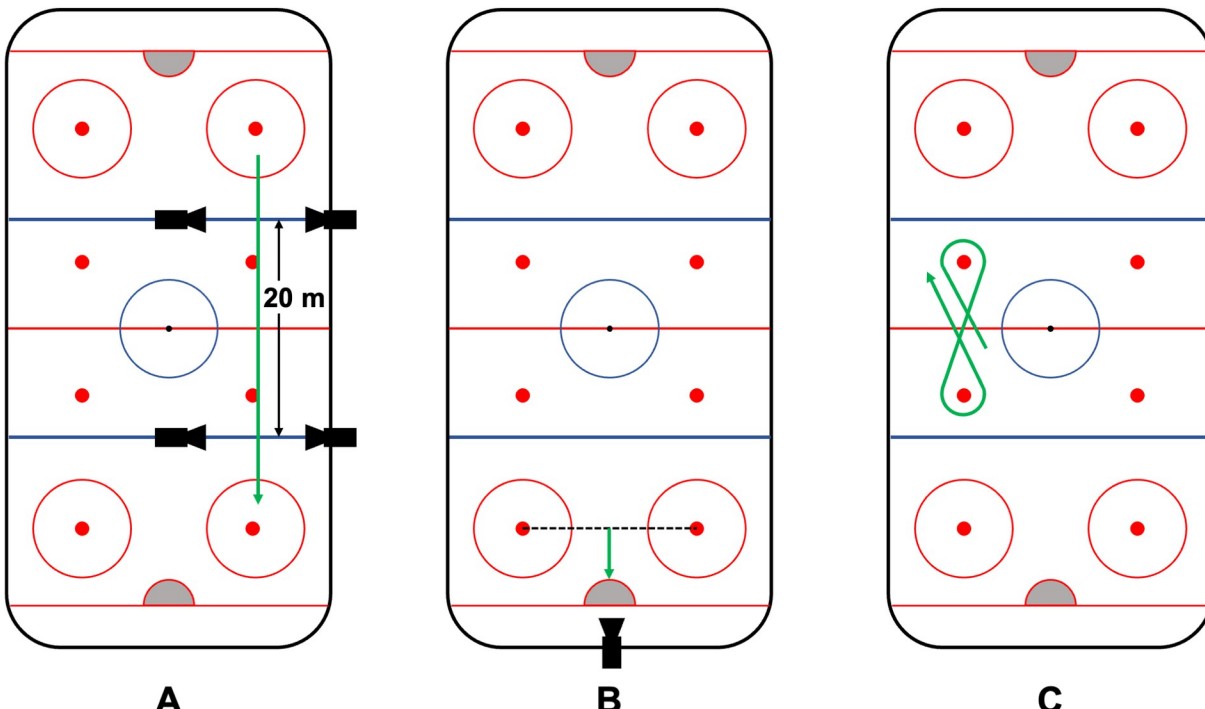

**Fig 1.** Illustration of the executed drills: (A) The sprint and the linear crossover were performed while skating down the long side of the rink with the timing lights positioned 20 m apart. (B) The setup for the slap shot, performed from a 5 m distance in front of the goal and the radar gun positioned behind the goal, and (C) the basic movement of the power turn, repeated three times on each side.

much rest as needed between the drills with a minimum of 30 seconds rest. The two tested conditions were with protective equipment (Equipment) and without protective equipment (No Equipment). In the Equipment condition, the players wore shin pads, shoulder pads, elbow pads, and the jock, while skates, helmet, gloves, and ice hockey stick were worn during both conditions.

Movement data were collected using the Xsens MVN Awina System (Xsens Technologies B.V., Enschede, Netherlands), a full-body system comprised of 17 inertial measurement units that were sampled at 60 Hz. Before testing, the wireless sensors were attached to a t-shirt accompanying the system or were attached directly to the participants' bodies using elastic straps. Additional clothing or equipment was worn over the sensors. In addition, sensors for the head and the feet were taped to the participants' helmet and skates, respectively. A calibration procedure (neutral standing and walking trial) was performed as per the manufacturer's recommendation before each of the three conditions.

The integrated sensors of the MVN Awina System capture the movements by recording the readings of the integrated accelerometer, the gyroscope, and the magnetometer. Using a biomechanical model based on the height and foot length of the participant, kinematics including the 3D position and linear and angular accelerations of all 23 segments are calculated. Additionally, the center of mass position and 3D joint angles of 22 joints can be exported from the system. For easier movement detection (see "movement Extraction" section), we reset the position of the segments by identifying the right heel as the origin (0,0,0) using the "reprocess" function of MVN Awinda software (2022.0). The raw joint angle and segment kinematic data are then exported. The three axes of interest are: the positive x-axis is anterior, the positive y-axis points to the left, and the positive z-axis is upwards [26].

## Analysis

**Movement extraction.** A set of movements were defined and extracted for each of the drills. The same number of individual movement trials using identical thresholds and sensors were extracted for all participants. For the sprint, individual strides were defined as the movement occurring between two ice contacts of the same foot. Strides were extracted using the lowpass filtered (zero-lag fourth-order Butterworth filter, cut-off frequency 5 Hz) velocity of the foot in the vertical direction to find the ice contact of the skate (Fig 2A). For each trial, three strides were extracted for each foot, providing six strides per trial. Therefore, in total 24 strides were available for 17 out of the collected 20 participants. One participant (S01) did not perform the sprint drill correctly, and the sensor data were compromised for two other participants (S04, S20).

For the linear crossover movement, one crossover was defined as the movement occurring between two ice contacts of the same foot. The ice contact detection was performed in the same way as it was performed for the sprint stride, using the lowpass filtered (fourth-order zero-lag Butterworth filter, cut-off frequency 5 Hz) velocity of the foot in the vertical direction. For each trial, one crossover movement was extracted for each foot, providing two crossover movements per trial (Fig 2B). Therefore, in total 8 crossovers were available for 17 out of the collected 20 participants. One participant (S01) did not perform the sprint drill correctly, and the sensor data were compromised for one other participant (S20). For one participant (S19) the needed number of crossover movements could not be extracted.

For the shooting trials, one shot per trial was performed and needed to be extracted. The shot was defined as the movement occurring between the start of the deflection of the hand

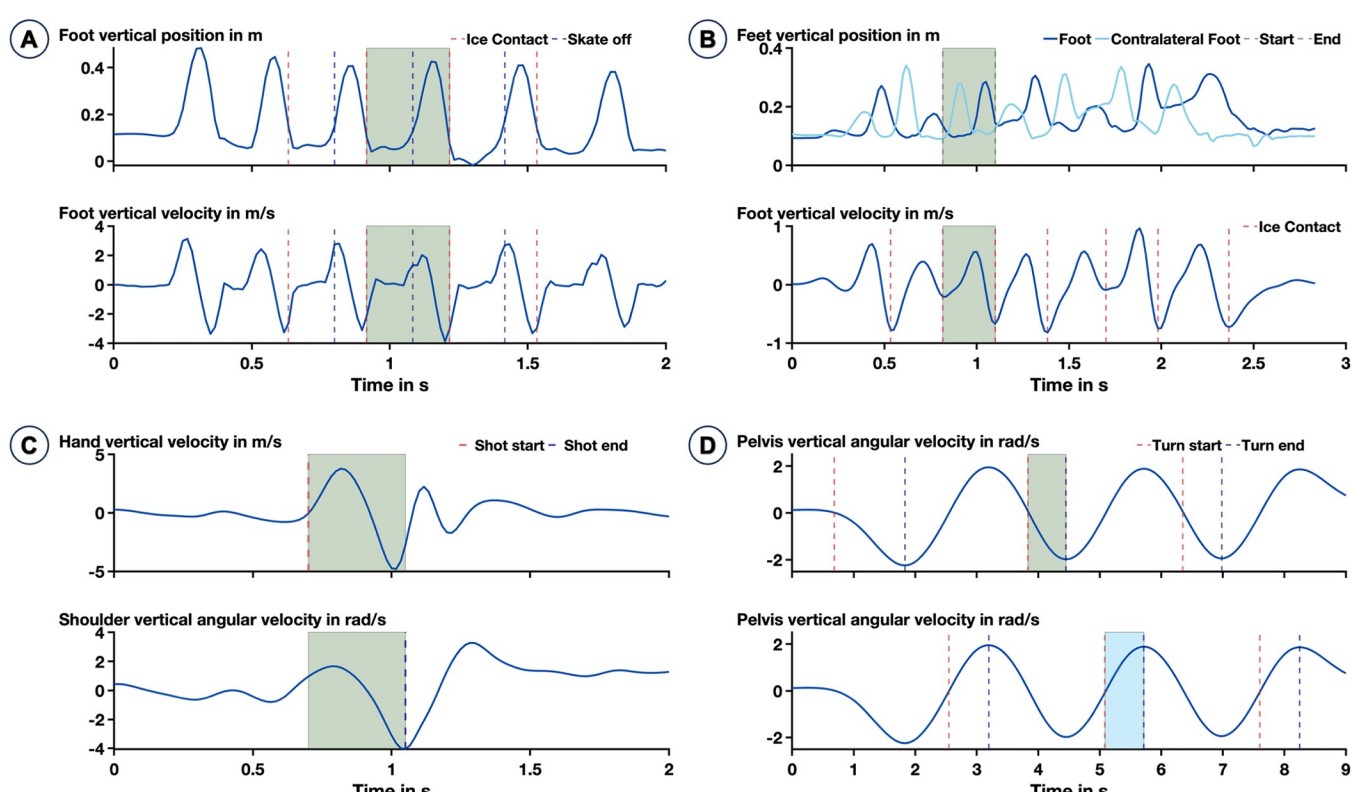

**Fig 2.** Movement extraction based on the sensor data for (A) the sprint stride, (B) the linear crossover movement, (C) the slap shot, and (D) the power turn. The shaded areas indicate one extracted movement. For the power turn, the upper plot shows a right turn while the lower plot is showing a left turn.

and the point in time when the stick hit the puck. Using the lowpass filtered (fourth-order zero-lag Butterworth filter, cut-off frequency 4 Hz) angular velocity of one shoulder in the vertical direction, the side of the shot was determined. Depending on the orientation, the respective angular velocity of the shoulder and the velocity of the hand in the vertical direction were used to determine the start and end of the shot (Fig 2C). Each participant performed four shots, which could be extracted for 18 out of the 20 participants. Two participants had corrupted sensor data and were excluded (S03, S20).

The power turn was separated into a sequence of two turns, one right-sided (clockwise) and one left-sided (counter-clockwise). For isolation of the single turns, the movement was split every time the lowpass-filtered (fourth-order Butterworth filter, cut-off frequency of 0.5 Hz) angular velocity of the pelvis in the vertical direction changed its sign (Fig 2D). In total 24 turns were available for 16 out of the collected 20 participants. One participant (S05) did not finish all the power turns completely so the required number of turns could not be extracted. Three other participants were not included in the analysis due to impaired sensor data (S01, S03, S20).

## Preprocessing

The 3D joint angles were the variables of interest for evaluating how equipment restricts the range of motion during ice hockey. For evaluation, the ankle, knee, hip, shoulder, elbow, and wrist were considered in all three rotational degrees of freedom. The extraction order follows the convention Z (flexion/extension), X (abduction/adduction), and Y (internal/external rotation) [27]. These conventions differ for some of the joints, e.g., pronation/supination for the elbow and wrist. One limitation of the MVN Awina system is its accuracy. Due to the limited accuracy of the system (1-degree root mean square error [26]), joint angles with maximal differences between the two conditions smaller than 1 degree were not considered reliable. Therefore, the wrist joint, ankle abduction/adduction, and knee abduction/adduction were excluded.

Only the ipsilateral joint angle trajectories were extracted for the sprint drill due to the symmetry of the movement. Due to the non-symmetry of the other three drills, the joint angles from both sides of the body were extracted for further analysis. The extracted trajectories were time-normalized to 100 time points using spline interpolation. Further, the joint angles were normalized participant-wise to a range between 1 and -1, which improves the efficiency and accuracy of the neural network (see section 2.3.3). To preserve the relative dependence in magnitude between the joints, the joint angles were all normalized to a pre-set ratio based on literature (S1 Table) [28]. Finally, all trajectories for one movement were concatenated into one single feature vector. For the sprint movement, the feature vector had the dimension of 1 x 1200 (12 trajectories and 100 samples) and for the remaining movements, the dimensions were 1 x 2400 (24 trajectories and 100 samples).

## Neural network

A shallow neural network was designed that consists of one input, one hidden, and one output layer for each drill. The number of nodes for the input layer was derived from the shape of the input data (12/24 joint angles, each with 100 time points), and the number of nodes in the output layer corresponded to the number of classifications results (two, i.e., Equipment and No Equipment). For the hidden layer, twice as many nodes as in the input layer were chosen and the hyperbolic tangent was used as an activation function. The architecture resulted in neural networks with 1200, 2400, and 2 nodes when one body side was considered, and neural networks with 2400, 4800, and 2 nodes, when both body sides were investigated. This three-layer

architecture was chosen since it is known that a single hidden layer is sufficient for learning most functional relationships [29]. Furthermore, this structure has shown good classification results in classifying running patterns using similar input data [30].

The network was trained on the entire dataset leaving one participant out for each of the different drills, and the validation during training was performed on randomly selected subsets. The best trade-off between training efficiency and model accuracy was found to be a mini-batch size of 16 and a maximum epoch size of 100. Due to the reduced amount of data for the shooting drill, the epoch size for the shooting was 50, while the remaining parameters were kept the same. The learning rate was set to 0.01.

The network's performance was determined by classifying one complete dataset of the one unseen participant that was not used for training. The accuracy per participant $p$ $Accuracy(p) = \frac{n_p}{N_p} \cdot 100$ can be calculated using the correctly classified movements $n$ and the total number of movements $N$. For the overall evaluation of the neural network, the single accuracies per participant were averaged for each movement. For example, for the sprint drill, the neural network was trained 17 times on 16 participants and tested on the remaining participant. The achieved testing accuracies were averaged across all 17 trials.

After training and testing the neural network participant-wise, layer-wise relevance propagation [22] was applied to the correctly classified trials to obtain the relevance scores for each input variable. The relevance scores indicate how relevant each variable was to the model's prediction. The resulting relevance pattern, therefore, consisted of 1200 and 2400 relevance scores, depending on the size of the input layer. Since the neighboring data points in the input signals were time-dependent and represented related information, each relevance pattern was smoothed to reduce variability in the signal without influencing the original pattern. The filter weighted the current point with 50% of its relevance and neighboring points with 25% of its relevance scores and was repeated twice. The weights for smoothing were chosen to sum to 1, therefore a repetitive application would mimic a Gaussian filter [24]. The participant-wise relevance scores were normalized. Finally, the normalized scores were averaged over all participants and rescaled to values ranging between zero and one, while values close to zero indicate the lowest relevance and values close to one highest relevance. All analysis was performed using MATLAB in the R2021b version (The MathWorks Inc., Natick, Massachusetts, USA) and the layer-wise relevance propagation toolbox by Bach and colleagues [22] was used for the implementation of the neural networks and all analyses based on layer-wise relevance propagation. The visualization and related processing steps of the results are based on code published by Hoitz et al. [24]. The supporting information of this work (S1 File) contains the data files required to replicate the results presented in this work.

## Statistical methods

The condition (Equipment vs. No Equipment) was considered as the independent variable and performance variables (sprint time, linear crossover time, slap shot puck velocity) were set for statistical analysis. For each participant, the times and velocities were averaged per drill and condition. Shapiro-Wilk test of normality was performed on the data and then, a paired two-sample Student's t-Test ($\alpha = 0.05$) was performed on the performance data.

## Results

### Performance

The performance measures were improved in the condition without equipment compared to the equipment condition. On average, the participants were able to perform the sprint 1.64%

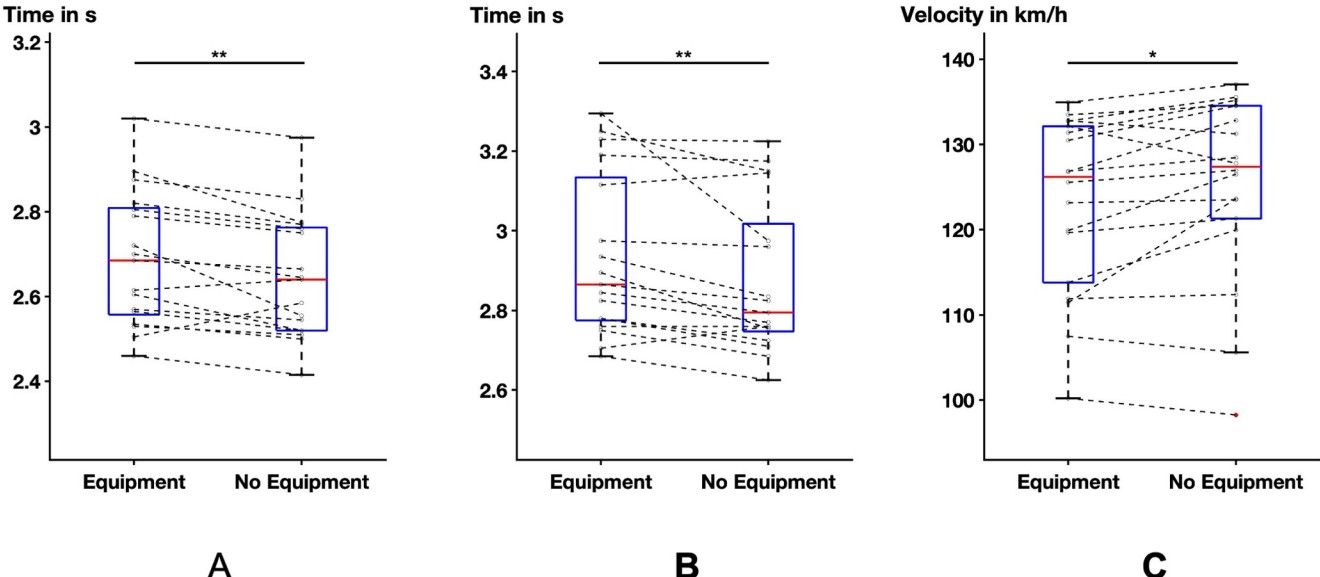

**Fig 3.** Boxplots of the performance time in seconds for (A) sprint and (B) linear crossover and the puck velocity in kilometers per hour for the slap shot for the two conditions: The central red line indicates the median and the bottom and top line indicate the 25th and 75th percentile, respectively. The connecting lines display the performance of the individual participants (* = p≤ 0.05, ** = p≤ 0.01).

faster without equipment compared to wearing the equipment (p = 0.0037, mean Equipment 2.72 s, mean No equipment 2.68 s). Two of the participants decreased their performance while 15 participants increased their performance without equipment (Fig 3A).

For the linear crossover, the participants performed the movement 2.65% faster on average in the no equipment condition (p = 0.0096, mean Equipment 3.03 s, mean No Equipment 2.94 s). Two participants increased the time they needed to perform the linear crossover, one participant needed the same amount of time, and 14 participants decreased the time needed (Fig 3B).

For the slap shot, the velocity of the puck decreased by 1.67% on average when the shooting was performed with equipment compared to the condition without equipment (p = 0.0274, mean Equipment 122.1 m/s, mean No Equipment 124.3 m/s). The velocity of the puck without equipment decreased for four participants, stayed the same for one participant, and increased for 14 participants (Fig 3C).

## Neural network

For each drill, the neural network was trained according to the number of participants, leaving one participant out. The accuracies across all participants were averaged for the classification results listed in Table 1.

**Table 1. Number of participants, number of samples, and best-achieved accuracy for each of the four drills.**

| Drill | Number of Participant | Number of samples | | Best Accuracy |
|---|---|---|---|---|
| | | Per participant | In total | |
| Sprint | 17 | 24 | 408 | 99.26% |
| Linear-Crossover | 17 | 8 | 136 | 97.79% |
| Slap Shot | 18 | 4 | 72 | 100% |
| Power Turn | 16 | 24 | 384 | 97.92% |

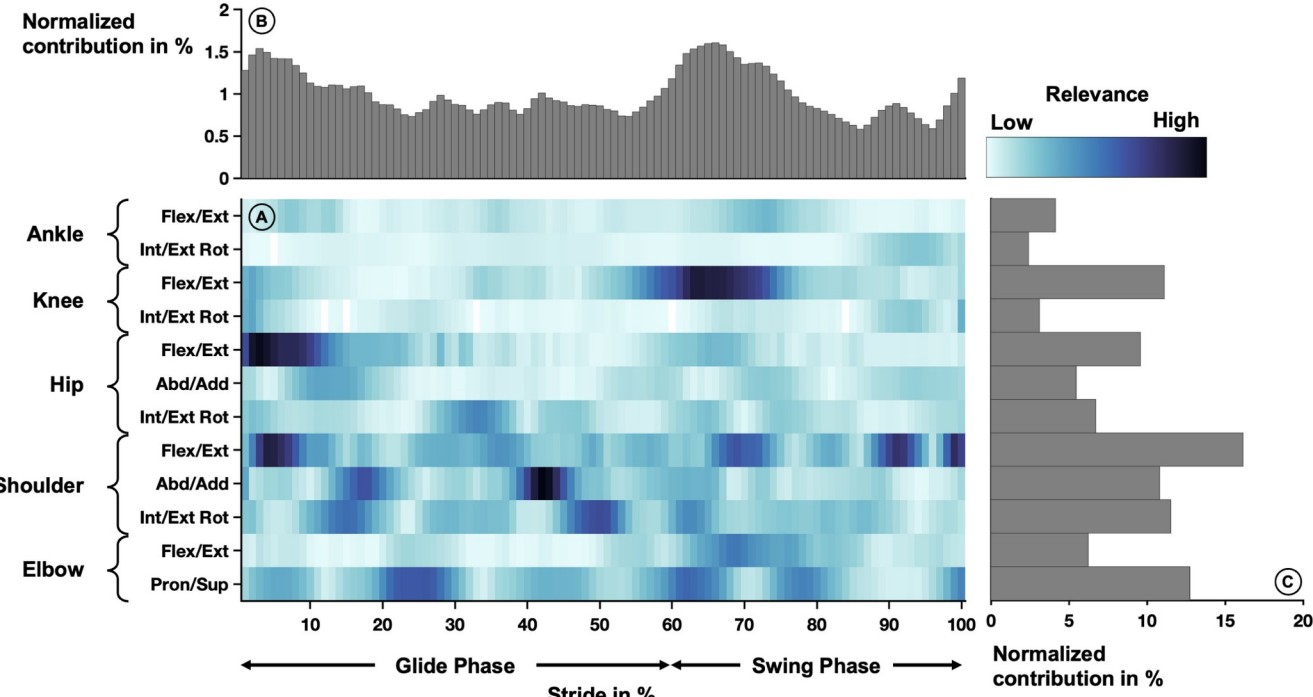

**Fig 4.** Averaged relevance scores per variable (A) over one sprint stride cycle and contributions within a step cycle (B) and joint angle trajectories (C). In the center (A), the relevance scores per joint angle over one sprint stride cycle are depicted by the shades of the color. The stride cycle starts with the ice contact of the foot and can be divided into the gliding phase and the swing phase indicated below the plot. Lighter colors indicate lower relevance and darker colors indicate higher relevance. The joint angles are listed on the left-hand side of the heatmap. Each row corresponds to one rotational degree of freedom. The top part of the figure (B) shows the vertical summation of the heatmap, highlighting the contribution of each percent of the stride cycle to the success of the model. The right side of the figure (C) depicts the horizontal summation of the heatmap, highlighting the contribution of each joint angle trajectory.

## Layer-wise relevance propagation

**Sprint.** The average relevance scores that were derived from the neural network model based on the correctly classified steps are depicted in Fig 4. Notable high relevance (Fig 4A) included knee flexion/extension at 60–75%, hip flexion/ extension at 0–10%, shoulder flexion/ extension at 0–10%, and shoulder abduction/adduction at 40–45%.

The variables recorded at each percent of the stride cycle were relevant for the classification of the equipment condition (Fig 4B). However, in the early swing phase after the skate-off (60–80%), a peak of high contributions was visible. Another peak of high relevance was located around the ice contact, starting at the end of the swing cycle (95–100%) and after the ice contact (0–10%).

The variables of trajectories that were most relevant were flexion/extension movements in the knee, hip, and shoulder. The joint trajectories related to the upper extremities (shoulder and elbow) showed a higher contribution to the classification result than trajectories related to the lower extremities. The lowest contributions were visible in all planes of the ankle and the knee and hip abduction/adduction and internal/external rotation (Fig 4C).

**Linear crossover.** The average relevance scores that were derived from the neural network model based on the correctly classified crossovers are depicted in Fig 5. Notable high relevance (Fig 5A) of the following leg included shoulder flexion/extension at 0–20% and shoulder internal/external rotation at 30–40%. For the crossing leg, notable high relevance included shoulder abduction/adduction at 95–100%, shoulder internal/external rotation at 85–95%, elbow flexion/extension at 70–85%, and elbow pronation/supination at 0–10%. In general, the clusters were associated with the body side in which the foot was in contact with the ice.

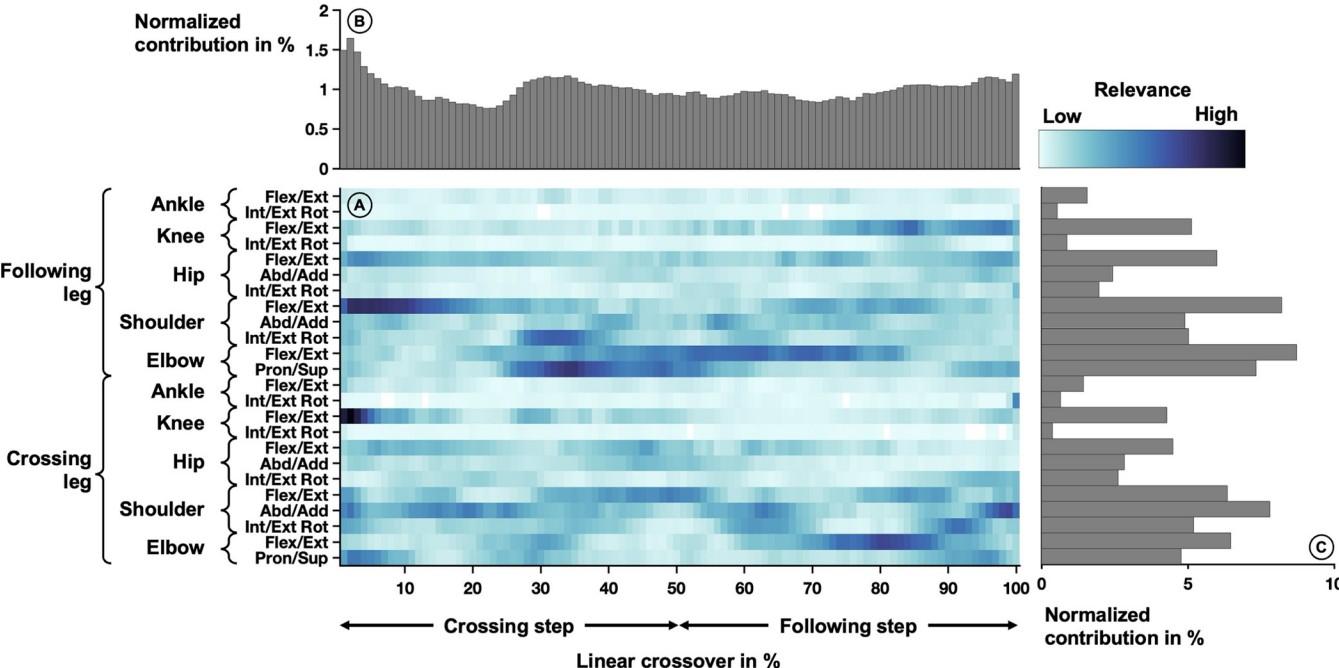

**Fig 5.** Averaged relevance scores per variable (A) over one linear crossover movement and contributions within a crossover cycle (B) and joint angle trajectories (C). In the center (A), the relevance scores per joint angle over one linear crossover movement are depicted by the shades of the color. The crossover movement starts with the ice contact of one foot and can be divided into the crossing step (contralateral foot) and the following step (ipsilateral side) indicated below the plot. Another division/naming would include the gliding phase and swing phase of the non-crossing foot. Lighter colors indicate lower relevance and darker colors indicate higher relevance. The joint angles are listed on the left-hand side of the heatmap. Each row corresponds to one rotational degree of freedom. The top part of the figure (B) shows the vertical summation of the heatmap, highlighting the contribution of each percent of the linear crossover movement to the success of the model. The right side of the figure (C) depicts the horizontal summation of the heatmap, highlighting the contribution of each joint angle trajectory.

The variables recorded at each percent of the linear crossover movement were relevant for the classification of the equipment condition and the relevance is roughly constant over the movement cycle (Fig 5B). However, at the beginning of the crossover movement (0–10%) and right before the touchdown (90–100%), a peak of high contributions was visible.

The variables of joint angle trajectories that were most relevant were related to the upper extremities (shoulder and elbow), with a slightly higher emphasis on the following leg. The relevance distributions in the lower extremities were almost symmetrical, while the flexion/extension movements in the knee and hip showed the highest contributions to the classification result. The lowest contributions were visible in all planes of the ankle, internal/external rotation of the knee and hip abduction/adduction, and internal/external rotation (Fig 5C).

**Slap shot.** The average relevance scores that were derived from the neural network model based on the correctly classified shots are depicted in Fig 6. Notable high relevance (Fig 6A) included elbow pronation/ supination at 20–25%, 80–90%, and 94–100%, and 80–90% shoulder flexion/extension.

The variables recorded at each percent of the slap shot were relevant for the classification of the equipment condition. (Fig 6B). Especially during the loading phase during which the arm is deflected to gain momentum, the relevance scores were almost constant with a slight tendency to decrease (0–55%). Starting around 55% the relevance scores increased and peaks were visible. Peaks were located at 73%, 84%, and 97%, directly before hitting the puck and completing the shot.

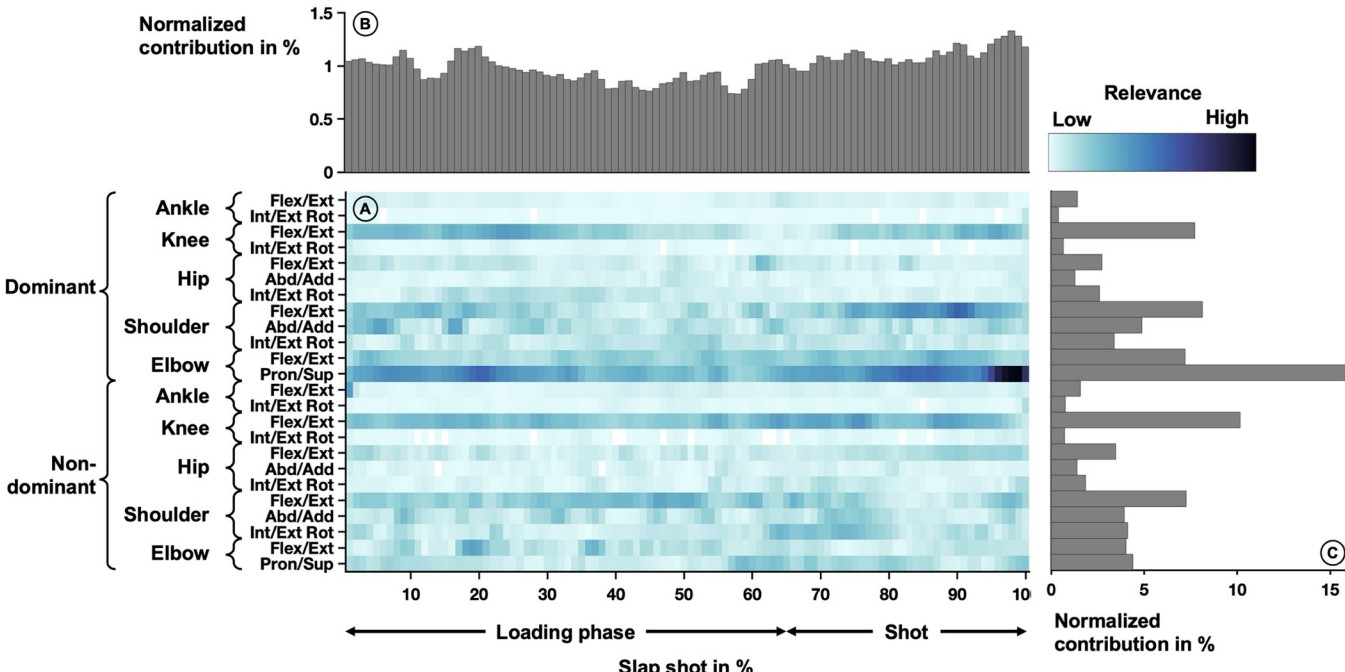

**Fig 6.** Averaged relevance scores per variable (A) over one slap shot movement and contributions within a slap shot cycle (B) and joint angle trajectories (C). In the center (A), the relevance scores per joint angle over one slap shot movement are depicted by the shades of the color The shooting movement starts with the initial movement of the hand and ends with striking the puck and can be divided into the loading phase (gain momentum) and the shooting phase. Lighter colors indicate lower relevance and darker colors indicate higher relevance. The joint angles are listed on the left-hand side of the heatmap. Each row corresponds to one rotational degree of freedom. The top part of the figure (B) shows the vertical summation of the heatmap, highlighting the contribution of each percent of the slap shot to the success of the model. The right side of the figure (C) depicts the horizontal summation of the heatmap, highlighting the contribution of each joint angle trajectory.

The most relevant joint trajectory was the pronation/supination movement of the elbow of the dominant side performing the shot. The highest relevance amongst the variables of the lower extremities was visible in the knee flexion/extension movement on both sides. The shoulder flexion/extension and elbow flexion/extension of the dominant side showed higher relevance in the upper extremities on the dominant side, while all variables related to the upper extremities showed similar relevance on the non-dominant side. The lowest relevance was visible in joint angles related to the ankle and hop and internal/external rotation movements of the knee. In general, the relevance scores are higher in the upper extremities. While the relevance scores of the lower extremities were symmetric on both sides, the relevance distributions of the upper extremities were different on both sides with a higher emphasis on the dominant arm (Fig 6C).

**Power turn.** The average relevance scores that were derived from the neural network model based on the correctly classified turns are depicted in Fig 7. Notable high-relevance clusters (Fig 7A) include flexion/extension around 45–55% of the inside-facing shoulder. Other clusters at this shoulder were located at 5–10% and 45–55% abduction/ adduction. The contralateral shoulder (facing outside) showed higher contribution at 0–10% and 45–50% flexion/extension.

The variables recorded at each percent of the stride cycle were relevant for the classification of the equipment condition (Fig 7B). However, a peak of higher contribution was visible at the beginning (0–10%) and in the middle of the movement (45–55%). Starting at 70% of the movement, the relevance increases in waveforms until reaching a peak at 98%.

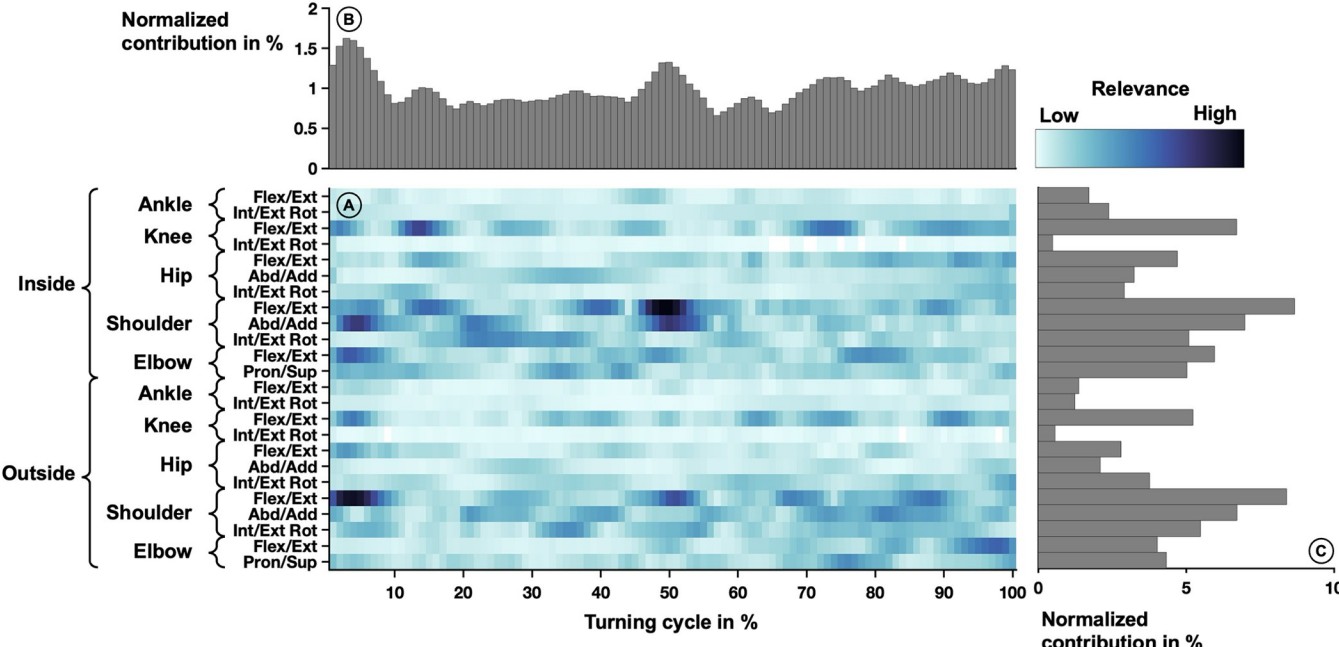

**Fig 7.** Averaged relevance scores per variable (A) over one power turn movement and contributions within a power turn cycle (B) and joint angle trajectories (C). In the center (A), the relevance scores per joint angle over one power turn movement are depicted by the shades of the color. Lighter colors indicate lower relevance and darker colors indicate higher relevance. The joint angles are listed on the left-hand side of the heatmap. Each row corresponds to one rotational degree of freedom. The top part of the figure (B) shows the vertical summation of the heatmap, highlighting the contribution of each percent of the power turn to the success of the model. The right side of the figure (C) depicts the horizontal summation of the heatmap, highlighting the contribution of each joint angle trajectory.

The variables of trajectories that were most relevant were joint trajectories related to the upper extremities (shoulder and elbow). In the lower extremities, flexion/extension in the knee and hip showed higher relevance, while the lowest contributions were visible in the ankle and abduction/ adduction and internal/external rotation movements of the hip and knee. In general, the relevance scores were higher in the upper extremities. The relevance scores of the inside-facing and outside-facing joint angles showed similar distributions, with a generally higher emphasis on the inside-facing side (Fig 7C).

## Discussion

The purpose of this study was to explore the effect of ice hockey protective equipment on player's performance and the possibility of detecting kinematic restrictions by an artificial neural network. Further, this study aimed to identify the variables and intervals during each of the movements that were most relevant in distinguishing between the two equipment conditions using layer-wise relevance propagation. The performance analysis showed that performance measures decreased significantly when the drills were performed with protective equipment. This outcome confirms previous findings that the use of protective equipment influences performance [8, 31, 32] which can partly be due to the added weight of the equipment and partly due to the restrictions incurred to the kinematics.

We demonstrated that an artificial neural network could accurately distinguish between the movements performed with and without equipment based on the changes in the kinematics. This suggests that there are distinct differences in the kinematics of the movement and the design of new protective equipment should include modifications to support more neutral

movements. However, only distinguishing the two conditions is not enough to make informed modifications on the protective equipment; Instead, knowing the kinematic variables and intervals that were most discriminative between the two conditions might lead to targeted changes in the protective equipment. Therefore, layer-wise relevance propagation was applied to highlight the relevance of each variable for the successful classification of the equipment conditions. The results showed (e.g., Fig 4) that not all variables contributed equally to the classification results, with some variables being more discriminative than others.

Comparing the most relevant variables of all four movements, similar variables showed high relevance scores across all four movements. The joints with the highest overall relevance scores were the shoulder joint in all three planes of movement (sagittal, frontal, and transverse) and the elbow joint in the investigated planes of movement (sagittal and frontal). Regarding the lower extremities, the sagittal plane (flexion/ extension) movements showed the highest relevance for the success of the model. The movements in other planes and movements related to the ankle showed low contributions across all four movements. An overview of the joints and planes of movement with the highest scores is given in Fig 8.

Comparing the relevance scores to the performed movements, the results were intuitive. For example, the finding, that the movements in the lower extremities (knee and hip) in the sagittal plane were more relevant than in the other two planes of movement. Knowing that the greatest range of motion is performed in the sagittal plane and therefore the most power is produced in this plane, the relevance scores fell in line with these expectations. Nevertheless, the relevance scores could provide a more detailed insight while depicting the relations between the different variables for deepening the understanding of movements performed under the two conditions.

By themselves, the relevance scores highlight which variables contributed the most to the classification result without answering the question of which consequences these findings have on the design of the protective equipment. Therefore, two assumptions were made: First, the assumption was made that the differences in performance result from the differences between the conditions. Further, it was assumed that the variables that were relevant to the neural network were the same variables that were relevant to the performance analysis. While

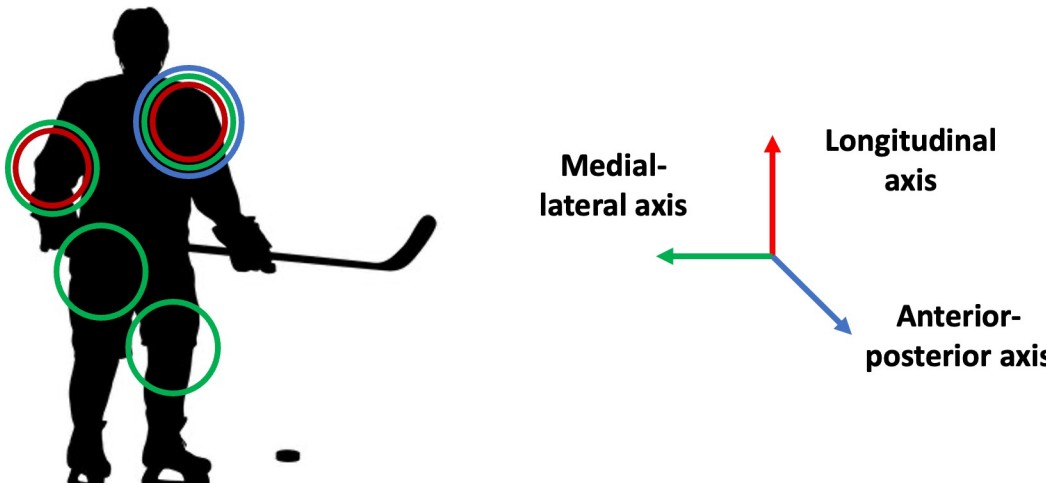

**Fig 8. The joints and planes of movement with the highest relevance across all four movements.** The different colored circles around the joints indicate the joint rotations that were most relevant across all four movements. Green circles are related to rotations along the medial-lateral axis (in the sagittal plane), blue circles are related to rotations along the anterior-posterior axis (in the frontal plane), and red circles to rotations along the longitudinal axis (in the horizontal plane).

performance was enhanced without equipment, the aforementioned assumption would need further evaluation.

Based on the assumptions, the relevance scores were combined with the corresponding joint trajectories, to link the relevant variables and the given performance results to make statements on how to improve the performance. As shown in S1 Fig, which depicts the shoulder flexion/extension movement of the sprint drill overlayed with the relevance scores, the relations between relevance scores and underlying joint angles could be made. In the mentioned example a higher extension of the shoulder joint in the no equipment condition in comparison to the equipment was visible. Furthermore, it has been shown that better performance is related to the condition without equipment. Therefore, the protective equipment should be changed in a way that the range of motion of the ice hockey player is closer to the natural movement pattern that was performed without the equipment. When applying this approach, concrete statements about improvements in protective equipment can be derived.

Each of the investigated movements requires a mix of power and finesse during execution to achieve maximum performance [33]. Having a closer look at the sprint drill, these improvements include increasing the flexion movement in the knee and the hip. This is important given that faster skaters show more flexion in the knees and hips compared to slow skaters [34, 35]. Increasing flexion to improve the power generated by the joints furthermore supports the need to improve the flexion movement in the elbow joint, as suggested by the results of the linear crossover and the power turns. Increasing the flexion in the shoulder is supported by the results of all four drills. Combining the results for the shoulder joint, it would be beneficial to design the shoulder pads in a way to allow for more shoulder flexion, higher adduction, and higher internal rotation. This finding is in line with previous findings. For example, a player who can perform smooth coordinated shoulder adduction/adduction movement is more efficient compared to players who cannot perform those movements [34]. Summarizing these findings, the ability of ice hockey players to move their arms closer to their trunks is improving their performance, especially during drills associated with turns and changes of direction such as the linear crossover and the power turn. This finding is supported by the physical effect that by reducing the moment of inertia, a higher angular acceleration can be achieved while having the same applied torque. In conclusion, the flexion movements should be increased in joints contributing to fast forward movements, while the protective equipment related to the upper body should be designed in a way that the arms can be brought closer to the body to support turning movements. Although some specific suggestions for enhancing the relevant ranges if certain joint motions, more detailed approaches will be needed to optimize the performance of each piece of equipment in isolation as well.

Ice hockey is a complex sport consisting of different movements that require a broad skill set of speed, agility, precision, and balance. The examined movements were chosen to address the different skills and to model different movement patterns. Therefore, contrasting results for one joint angle for different movements may not surprise. As an example, the results of the linear crossover suggest an increased pronation movement of the elbow, while in shooting a decreased pronation movement seemed beneficial. To understand the biomechanical mechanisms of the pronation/supination movement of the elbow more in-depth, further analysis is required. Moreover, a decision should be made depending on which movement has the higher prevalence during a game or according to the position of the hockey player.

In summary, this study's work showcased how artificial neural networks and layer-wise relevance propagation could be used to identify the variables that best capture how equipment influences movements. Using the variables of interest instead of the whole available dataset streamlined the process of selecting the right variables, removed the researcher-based selection bias, and made the process more efficient. Further, the given performance measures can be

used to derive improvement suggestions regarding the protective ice hockey equipment. Nevertheless, the results should be evaluated critically in the context of the given problem regarding the usability and the informative value of the results.

One limitation is related to the preselection and preprocessing of the data. First, some joint angles were not investigated since their magnitude ranged in the dimension of the measurement error of the MVN Awina system (1-degree root mean square error [26]). Secondly, a joint-dependent normalization was applied to maintain the relative information on the range of motion between the joints. From a computer science perspective, this normalization skewed the results in favor of the joints with larger ranges of motion, while a uniform normalization would have favored joints with a smaller range of motion such as knee internal/external rotation. The choice of using unequal normalization was made with regard to the practical application of this study, but it should be kept in mind that this normalization method introduced a biomechanical bias.

Finally, the neural network itself has some limitations that may influence the relevance scores. First, the limited number of training data for some of the movements, e.g., the slap shot, might lead to either low accuracy or limited generalization [36]. Further, based on their functionality, artificial neural networks might not find *the best* solution (global minimum) but rather *a good* solution (local minimum). Precautions were undertaken to avoid these problems, e.g., testing on unseen participants and testing different hyperparameter settings. Nevertheless, these limitations should be kept in mind during evaluation.

## Summary and conclusion

The potential of neural networks and layer-wise relevance propagation has not yet been explored to make functional statements in an applicable case. Thus, the purpose of this work was to apply this combination to explore and understand how complex drills in ice hockey were affected by the presence of equipment based on their joint angles captured by an inertial measurement unit-based system. Further, the variables and intervals during each of the movements that were most relevant in distinguishing between the two equipment conditions using layer-wise relevance propagation were identified.

In summary, the variables contributing the most to distinguishing between the Equipment and No Equipment conditions were related to the upper extremities and movements in the sagittal plane/along the medial-lateral movement axis. Using the highlighted variables by the proposed methodology and linking the results to the performance measures, suggestions on equipment improvements could be derived. Changes to the equipment should support the flexion movements of the knee and hip and should allow players to keep their upper extremities closer to the torso.

While the presented methodology has some limiting factors, their effect can be minimized by interpreting the results within the contextual information of the model, the underlying data, and influencing factors to translate the relevance scores into meaningful statements.

## Supporting information

**S1 Table. The 3-dimensional joint angles and their ranges of motion.** Ranges of motion are given in the full range of motion and split values in the two movement directions. The ratio was calculated by dividing the individual joint angles by the overall maximum range of motion (shoulder abduction/adduction) [28].
(PDF)

**S1 Fig. Shoulder joint flexion/extension trajectories for each condition overlayed with the relevance scores of the respective joint for one sprint stride.** The joint trajectories are displayed with the mean (solid line) ± standard error (shaded area) for the two conditions, equipment in red and no equipment in green over one sprint stride cycle. The vertical line indicates the skate-off and therefore the transition from the gliding to the swing phase of the foot. The plots were overlayed with the relevance scores calculated for the shoulder flexion/extension trajectory. Dark colors indicate higher relevance and lighter colors indicate lower relevance. (TIF)

**S1 File. Data.** This archive contains all the underlying data (raw and processed) presented in this publication.
(ZIP)

## Author Contributions

**Conceptualization:** Rebecca Lennartz, Arash Khassetarash, Sandro R. Nigg, Bjoern M. Eskofier, Benno M. Nigg.

**Data curation:** Rebecca Lennartz, Arash Khassetarash.

**Formal analysis:** Rebecca Lennartz.

**Funding acquisition:** Sandro R. Nigg.

**Investigation:** Rebecca Lennartz.

**Methodology:** Rebecca Lennartz.

**Supervision:** Bjoern M. Eskofier, Benno M. Nigg.

**Visualization:** Rebecca Lennartz.

**Writing – original draft:** Rebecca Lennartz, Arash Khassetarash.

**Writing – review & editing:** Rebecca Lennartz, Arash Khassetarash, Sandro R. Nigg, Bjoern M. Eskofier, Benno M. Nigg.

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
