## [Decision Letter · Decision Letter 0]

22 Jul 2024

PONE-D-24-06333Neural network and layer-wise relevance propagation reveal how ice hockey protective equipment restricts players' motionPLOS ONE

Dear Dr. Lennartz,

Thank you for submitting your manuscript to PLOS ONE. After careful consideration, we feel that it has merit but does not fully meet PLOS ONE’s publication criteria as it currently stands. Therefore, we invite you to submit a revised version of the manuscript that addresses the points raised during the review process.

We look forward to receiving your revised manuscript.

Kind regards,

Yaodong Gu

Academic Editor

PLOS ONE

Additional Editor Comments:

The currently methods shall be discussed to approved the validation.

Reviewers' comments:

Reviewer's Responses to Questions

**Comments to the Author**

1. Is the manuscript technically sound, and do the data support the conclusions?

Reviewer #1: Yes

Reviewer #2: Partly

2. Has the statistical analysis been performed appropriately and rigorously? 

Reviewer #1: Yes

Reviewer #2: No

3. Have the authors made all data underlying the findings in their manuscript fully available?

Reviewer #1: Yes

Reviewer #2: Yes

4. Is the manuscript presented in an intelligible fashion and written in standard English?

Reviewer #1: Yes

Reviewer #2: Yes

5. Review Comments to the Author

Reviewer #1: Review comment

This manuscript entitled “Neural network and layer-wise relevance propagation reveal how ice hockey protective equipment restricts players' motion” primarily aimed to quantify the effects of protective equipment on ice hockey players' performance and to identify the kinematic restrictions caused by the equipment using neural networks and layer-wise relevance propagation. The results of this study provide guidance for sport biomechanics and bioengineering. While it is a very interesting topic. But I think this manuscript has some flaws to fill in before it can be published in a journal. There are several questions should be addressed, which list below. This is a very significant study, and I suggest that it be published in this journal after revision.

Specific comments

1. In the abstract part, “A neural network accompanied by layer-wise relevance propagation was applied to the 3D kinematic data to identify restrictions incurred by the protective equipment.” How does layer-wise relevance propagation specifically contribute to identifying restrictions in kinematics.

2. In the Introduction part, “Conventional biomechanical analysis methods addressed the problem of high dimensionality data by focusing on single, time-discrete variables that are pre-selected by the researcher.” Please use more than one reference to support this sentence.

3. In the Methods part, “Twenty male ice hockey players performed four different drills with and without protective equipment while their performance was quantified.” What criteria were used to select the twenty participants.

4. In the Discussion part, “The performance analysis showed that performance measures decreased significantly when the drills were performed with protective equipment.” How might these findings influence future designs of ice hockey protective equipment.

Reviewer #2: Dear Authors,

The manuscript is very clear, and the problem addressed is of significant value. However, there are still some issues that need to be resolved.

Major Issues:

1.The manuscript mentions the use of the Xsens Awinda System for testing. Could you please clarify whether the data was exported using the Xsens Analysis software? If so, kindly specify the version of the software used in the manuscript.

2.If the Xsens Analysis software was utilized for data export, please indicate whether the recalculation function of the software was employed. Did you filter the data directly without using the recalculation function, or did you use this function? If the recalculation function was used, please provide details on the quality settings applied during recalculation.

3.The manuscript indicates that all data were used for training the neural network. This approach may lead to overfitting issues in the trained model. Please provide an explanation or justification for this approach and discuss any measures taken to mitigate potential overfitting.

Minor Issues:

1.If possible, please include photographs showing the placement of the Xsens sensors, particularly on the head and feet, as these sensors were positioned in unconventional locations according to the manuscript.

2.The cutoff frequency of the filters used for processing experimental data varied across different subjects. Could you explain the rationale behind these changes? Were these adjustments based on the results of spectral analysis or guided by previous literature?

6. PLOS authors have the option to publish the peer review history of their article (what does this mean?). If published, this will include your full peer review and any attached files.

Reviewer #1: **Yes: **Zixiang Gao

Reviewer #2: No

---

## [Author Response · Author response to Decision Letter 0]

4 Sep 2024

Dear reviewers, 

thank you for your constructive comments on our manuscript. Below we respond to each of the comments for both reviewers. Responses are indicated in Italic font.

Further changes included smaller linguistic revisions and switching from Canadian to American English. 

Thank you.

Reviewer #1: 

This manuscript entitled “Neural network and layer-wise relevance propagation reveal how ice hockey protective equipment restricts players' motion” primarily aimed to quantify the effects of protective equipment on ice hockey players' performance and to identify the kinematic restrictions caused by the equipment using neural networks and layer-wise relevance propagation. The results of this study provide guidance for sport biomechanics and bioengineering. While it is a very interesting topic. But I think this manuscript has some flaws to fill in before it can be published in a journal. There are several questions should be addressed, which list below. This is a very significant study, and I suggest that it be published in this journal after revision.

Specific comments:

1.In the abstract part, “A neural network accompanied by layer-wise relevance propagation was applied to the 3D kinematic data to identify restrictions incurred by the protective equipment.” How does layer-wise relevance propagation specifically contribute to identifying restrictions in kinematics.

Thank you for pointing that out. We rewrote the sentences in lines 28-30 and added the assumption, that the detected changes in the kinematics are due to the restrictions incurred by the protective equipment.

2. In the Introduction part, “Conventional biomechanical analysis methods addressed the problem of high dimensionality data by focusing on single, time-discrete variables that are pre-selected by the researcher.” Please use more than one reference to support this sentence.

We added examples and supported the statements with references in line 68.

3. In the Methods part, “Twenty male ice hockey players performed four different drills with and without protective equipment while their performance was quantified.” What criteria were used to select the twenty participants.

All participants had to be male active ice hockey players. No further criteria were applied. We changed the description in lines 117 - 119 to make it clearer. Additionally, we added an explanation of why we focused on male participants. 

4. In the Discussion part, “The performance analysis showed that performance measures decreased significantly when the drills were performed with protective equipment.” How might these findings influence future designs of ice hockey protective equipment.

Thank you for your comments. The results of our analysis provide specific clues for future design. Our focus was to provide a methodology to identify the most relevant joints influencing the range of motion. While we have made specific suggestions for enhancing the relevant ranges of certain joint motions, we acknowledge that more detailed approaches will be needed to optimize the performance of each piece of equipment in isolation as well. We clarified that in lines 557-559.

Reviewer #2: 

Dear Authors,

The manuscript is very clear, and the problem addressed is of significant value. However, there are still some issues that need to be resolved.

Major Issues:

1. The manuscript mentions the use of the Xsens Awinda System for testing. Could you please clarify whether the data was exported using the Xsens Analysis software? If so, kindly specify the version of the software used in the manuscript.

We specified the software and its version used for exporting the data in the manuscript in lines 170-173.

2.If the Xsens Analysis software was utilized for data export, please indicate whether the recalculation function of the software was employed. Did you filter the data directly without using the recalculation function, or did you use this function? If the recalculation function was used, please provide details on the quality settings applied during recalculation.

We added the information in lines 170-173. We used the “reprocess” function to identify the right heel as the origin (0,0,0). This helped us to have consistency in movement detection algorithms. We exported the raw joint angles and then used MATLAB for filtering as needed, e.g., for the movement extraction. The joint angles used for the classification were not filtered. Only time-normalization and magnitude-normalization were performed as described in lines 235-240. 

3.The manuscript indicates that all data were used for training the neural network. This approach may lead to overfitting issues in the trained model. Please provide an explanation or justification for this approach and discuss any measures taken to mitigate potential overfitting.

Thank you for pointing that out. We used the leave-one-participant-out approach and had additional information in the Results – Neural Network part. We moved the explanation up into the Methods – Neural Network part (lines 256, 266-268) and shortened the explanation in lines 324-325.

Minor Issues:

1. If possible, please include photographs showing the placement of the Xsens sensors, particularly on the head and feet, as these sensors were positioned in unconventional locations according to the manuscript.

Thank you for your comment. Unfortunately, we do not have photographs of the placement of the sensors. Further, due to a change of institutions since the data collection, we are not able to recreate the positions of the sensors either. 

2.The cutoff frequency of the filters used for processing experimental data varied across different subjects. Could you explain the rationale behind these changes? Were these adjustments based on the results of spectral analysis or guided by previous literature?

Thank you for your comment. The cutoff frequency did not vary between subjects. Since we used different data streams and therefore different features for the four drills, we had different cutoff frequencies between drills/movements. Therefore, no further analysis in this direction was performed.

---

## [Decision Letter · Decision Letter 1]

4 Oct 2024

Neural network and layer-wise relevance propagation reveal how ice hockey protective equipment restricts players' motion

PONE-D-24-06333R1

Dear Dr. Lennartz,

We’re pleased to inform you that your manuscript has been judged scientifically suitable for publication and will be formally accepted for publication once it meets all outstanding technical requirements.

Kind regards,

Yaodong Gu

Academic Editor

PLOS ONE

Additional Editor Comments (optional):

Well done!

Reviewers' comments:

Reviewer's Responses to Questions

**Comments to the Author**

1. If the authors have adequately addressed your comments raised in a previous round of review and you feel that this manuscript is now acceptable for publication, you may indicate that here to bypass the “Comments to the Author” section, enter your conflict of interest statement in the “Confidential to Editor” section, and submit your "Accept" recommendation.

Reviewer #2: All comments have been addressed

2. Is the manuscript technically sound, and do the data support the conclusions?

Reviewer #2: Yes

3. Has the statistical analysis been performed appropriately and rigorously? 

Reviewer #2: Yes

4. Have the authors made all data underlying the findings in their manuscript fully available?

Reviewer #2: Yes

5. Is the manuscript presented in an intelligible fashion and written in standard English?

Reviewer #2: Yes

6. Review Comments to the Author

Reviewer #2: (No Response)

7. PLOS authors have the option to publish the peer review history of their article (what does this mean?). If published, this will include your full peer review and any attached files.

Reviewer #2: No

---

## [Editor Report · Acceptance letter]

7 Oct 2024

PONE-D-24-06333R1 

PLOS ONE

Dear Dr. Lennartz, 

I'm pleased to inform you that your manuscript has been deemed suitable for publication in PLOS ONE. Congratulations! Your manuscript is now being handed over to our production team.

Kind regards, 

on behalf of

Professor Yaodong Gu 

Academic Editor

PLOS ONE